# Comparing the hamstring muscle activity between injured and non-injured sides during a variety of Nordic hamstring exercises

Taspol Keerasomboon[1©], Parunchaya Jamkrajang[1], Weerawat Limroongreungrat[1], Thammanunt Chrunarm[1], Toshiaki Soga[2,3], Norikazu Hirose[4©]*

1 College of Sports Science and Technology, Mahidol University, Bangkok, Thailand, 2 Graduate School of Engineering and Science, Shibaura Institute of Technology, Tokyo, Japan, 3 Research Fellow of Japan Society for the Promotion of Science, Japan, 4 Faculty of Sport Sciences, Waseda University, Tokyo, Japan

© These authors contributed equally to this work.
* toitsu_hirose@waseda.jp

## Abstract

### Objectives

To compare the electromyographic activity of the Biceps femoris long head and Semitendinosus muscles during bilateral and unilateral isometric Nordic hamstring exercise performed on an inclined platform at different knee flexion angles between leg side with and without a history of hamstring strain injury.

### Methods

Ten physically active male volunteers with a history of hamstring strain injury in either leg performed isometric Nordic hamstring exercise, maintaining the exercise position for 5 s under the following conditions: (1) bilateral, 150° knee angle on a 50° inclined platform (BL50); (2) bilateral, 140° knee angle on a 40° inclined platform (BL40); (3) unilateral, 150° knee angle on a 50° inclined platform (UL50); and (4) unilateral, 140° knee angle on a 40° inclined platform (UL40). External torque on the knee joint was controlled to ensure equivalence across conditions. Electromyographic activity of the Biceps femoris long head, Semitendinosus, and related muscles was measured in both legs with and without a history of hamstring strain injury.

### Results

The Biceps femoris long head muscle demonstrated significantly higher electromyographic activity during the unilateral Nordic hamstring exercise tasks, irrespective of hamstring strain injury history (p < 0.05). The Biceps femoris long head electromyographic activity was higher than Semitendinosus activity only in unilateral Nordic hamstring exercise conditions.. Additionally, no significant differences in the

**Data availability statement:** All relevant data are within the manuscript and its Supporting Information files.

**Funding:** The author(s) received no specific funding for this work.

**Competing interests:** No authors have competing interests.

electromyographic activity were observed across the different slope angles when the external torque at the knee joint was matched ($p > 0.05$).

## Conclusions

These findings suggest that performing isometric unilateral Nordic hamstring exercise at shallow knee flexion angles preferentially enhances the muscle activation of biceps femoris long head muscle regardless of hamstring strain injury history.

## Introduction

Hamstring strain injury (HSI) is a common injury in sports that involve sprinting-related activities, such as track and field events [1], soccer [2], and rugby [3]. Additionally, HSI not only has a remarkable occurrence rate but also exhibits a substantial recurrence rate of approximately 35% [4]. Furthermore, approximately 50% of athletes who have previously suffered hamstring injuries experience recurrence within 25 days of resuming play [5]. According to prior studies, hamstring injuries increased by two folds from 12% to 24% between 2001–2022 [6]. After HSI injury, players usually return to full team training after approximately 20 days, and preventing the first hamstring injury and minimizing the likelihood of a recurrence is crucial to enhancing athletic performance in football [7]. Additionally, a history of HSI is a strong risk factor for future HSI injury which has been reported to increase the risk of the injury by as much as 11.6 fold in professional soccer [8] because of structural [reduced biceps femoris fascicle length [9], atrophy [10], scar tissue [11], and neurological (reduced voluntary activation) [12] maladaptation within the injured muscle, which may explain reduced hamstring strength [13] following a HSI. Persistent deficits may reduce the ability of the hamstrings to tolerate high stress and strain levels, contributing to an elevated risk of recurrence. Hence, avoiding initial and recurring HSI is imperative for enhancing athletic performance [14].

Most hamstring strain injuries are thought to occur during the late swing phase [15–20], and their occurrence rate during sprinting can reach 70% in soccer players [7] and 93% in sprinters [21]. Although it is widely assumed that there is an eccentric hamstring muscle fiber action during the swing phase of high-speed running, there may be an isometric contraction of the hamstrings because computational modeling data indicate that the contractile element (CE) passively increases during the early swing phase, which remains mostly constant when electromyographic (EMG) activity and forces are higher during the late swing phase [22,23] Furthermore, active elongation of the CE, as shown in computer modeling studies during the mid-swing phase, cannot be understood only as an eccentric muscle action because of the effect of muscle kinematics [22].

Sprint-type HSI is characterized by greater susceptibility of the biceps femoris long head (BFlh) muscle compared with the semitendinosus (ST) and semimembranosus (SM) muscles, which are other biarticular hamstring muscles [24,25]. The BFlh experiences a greater increase in muscle-tendon unit length during the swing phase of

high-speed running. Furthermore, the late swing phase is characterized by increased EMG activity of the BFlh muscle compared with earlier phases [26]. During the early stance phase, the EMG activity of the BFlh is also higher than that of the ST [27]. In addition, a recent study investigated muscle activation characteristics in soccer players with a history of HSI during sprinting. This study's findings indicated that individuals with HSI exhibit inadequate muscle activity, contributing to trunk stability [28]. This raises an interest in further investigating the differences in muscle activity between the history of HSI-injured and non-injured HSI. Although the key concept for preventing HSI has been widely accepted in the development of eccentric strength contractions in hamstring muscles [9,29–31] based on previous studies [22,23], isometric strength cannot be ignored. Additionally, isometric contractions are commonly preferred in the initial stages of rehabilitation to prevent injury from occurring in the first place [31]. Insufficient isometric strength, resulting in force eccentric lengthening during the late swing phase of high-speed running or braking during the same phase, is suggested as a possible factor contributing to injury [22,23]. Nevertheless, there is no consensus on the most effective approach for rehabilitating hamstring injuries [32], which may account for the extended recovery periods commonly encountered by athletes [33].

Although several preventive exercises were shown to diminish the occurrence of hamstring strain injuries (HSI) among professional and semi-professional football players, including eccentric exercises, the FIFA 11 + protocol, balance training, and core stability exercise [34], Nordic hamstring exercise (NHex) is frequently recommended to prevent HSI [35]. In contrast, some studies suggest that the NHex is inadequate [35–37] because of the greater ST muscle activation compared with the BFlh muscle, which is more susceptible to damage [36]. A recent study has shown that positioning at a shallow knee flexion angle during NHex increased the activation of the BFlh muscle [35]. However, previous studies could not provide a clear underlying mechanism for this result. One possible mechanism may be a change in hamstring muscle length during NHex at a shallow knee flexion angle. Additionally, a previous study showed that adjusting the slope of the lower leg support to 20° and 40° allowed the participants to perform the movement with a greater range of motion amplitude in a controlled manner, with peak joint torques attained at longer hamstring lengths [38]. Moreover, the NHex procedure can be performed unilaterally using an inclined platform with a shallow knee flexion angle that preferentially recruits the BFlh muscle [39]. However, the specific activation patterns of the BFlh and ST muscles during isometric NHex in a unilateral position are unclear. Hence, it would be interesting to investigate the activities of the BFlh and ST muscles during the isometric unilateral NHex when the external torque at the knee joint is equivalent. This study aimed to investigate the activities of the BFlh, ST, and related muscles during isometric unilateral NHex with an inclined platform at different knee angles when the external torque at the knee joint was equivalent in individuals with HSI to identify preferable NHex settings for HSI rehabilitation and prevention.

## Materials and methods

### Experimental approach to the problem

In this study, we used a crossover design to perform isometric NHex variations on an inclined platform. Firstly, to examine the hamstring EMG pattern between bilateral and unilateral NHex, there were two conditions of NHex: bilateral and unilateral NHex. To understand how changing hamstring length affects hamstring activity, there were two slopes, each with a different knee angle. The external torque at the knee joint was calculated to adjust the slope and knee angles to be equivalent to a knee flexion angle of 140° at a 40° slope (Fig 1). However, the knee joint angle is shallower in A than in B, and the hamstring muscles are more extended in A. Based on previous studies, BFlh may work more in A than B.

### Participants

The sample size in the present study was determined using G*Power 3.1.9.4 software (Heinrich Heine University, Dusseldorf, Dusseldorf, Germany). It was set as an ANOVA repeated measure within factors with a significance level of 0.05, effect size of 0.61, and power of 0.8. Effect size was calculated using partial $\eta^2 = 0.27$ based on a previous study [40]. As a

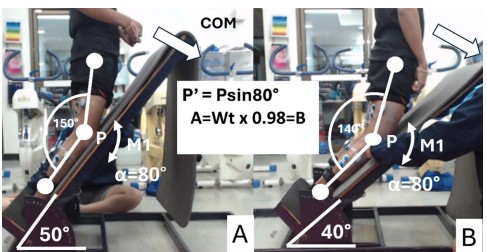

**Fig 1. showed how to calculate the external torque at knee joint between NHex A and B.** M1, External torque at knee joint will be equivalent between A and B which means the same intensity, COM, Center of Mass.Calculation of External Torque at Knee Joint Between NHex A and B.

result, we confirmed that a sample size of eight was sufficient. Therefore, to account for potential dropouts, 10 male active volunteers with a history of HSI within two years in either leg and performed aerobic activity at least twice a week (mean age, 20.9±1.2 years; height, 1.74±0.07 m; and body mass, 72.8±11.6 kg, all measured in mean±standard deviation [SD]) participated in this study. Participants carried out an injury history questionnaire that specified the location, severity, and rehabilitation duration of their most recent HSI and the accumulated number of hamstring strain injuries they had experienced. The recruitment process was carried out from the 6th of November 2023 to the 29th of November 2023. The exclusion criteria were as follows: patients with anterior cruciate ligament injury, a recent history of lower limb injury within 6 months prior to the experiment and the NHex was not previously performed. The study protocol was approved by the Institutional Review Board of Mahidol University (approval number: MU-CIRB 2023/226.1407), and all the procedures were performed in accordance with the Declaration of Helsinki. All the participants were informed of the purpose and procedure of the study, and written informed consent was obtained from all the participants before their participation in the study.

## Procedures

Before the experiment, the participants performed a warm-up and were prepared for surface EMG electrode placement by shaving the hair around the target site and cleaning the skin with alcohol. The electrodes were attached to the seven target muscles, namely the BFlh, ST, gluteus maximus (GM), rectus abdominis (RA), elector spinae (ES), lateral gastrocnemius (LG), and medial gastrocnemius (MG), and they were measured using circular bipolar surface electrodes in both legs (Ambu®, type Blue Sensor P-00-S/50, Ag/AgCl, diameter: 13 mm, center-to-center distance: 25 mm, Ballerup, Denmark). The electrodes were placed on each target muscle following the landmarks: midpoint between the ischial tuberosity and lateral epicondyle of the tibia (BFlh); midpoint of the line between the ischial tuberosity and medial epicondyle of the tibia (ST); on the line between the medial condyle of the tibia and ischial tuberosity (SM); midpoint between the sacral vertebrae and greater trochanter (GM); two finger width lateral from the spinous process of the L1 vertebra (ES), and two finger-width lateral from the midline of the umbilicus (AD); and midpoint on the most prominent bulge of the muscle between the proximal and posterior part of the medial condyle, adjacent to the femur, capsule of the knee joint, and middle part of the posterior surface of the calcaneus (MG), at 1/3 of the line between the head of the fibula and heel (LG). Electrodes were placed parallel to the lines between these landmarks as recommended by the SENIAM guidelines (Hermens et al., 2000). To achieve accurate electrode placement on each muscle, the examiner palpated the muscle bellies and tested the EMG signal. After EMG placement, the participants performed maximal voluntary isometric contractions (MVICs) to normalize the following muscles. For the BFlh and ST, the participants were instructed to perform two repetitions of prone leg curl at 45° and 90° of knee flexion with an MVIC using manual resistance. The reason why this study selected prone leg curl at 45° and 90° is because a previous study reported that leg curl with knee flexion angles between 30 and 60° activate BFlh, whereas ST was higher than that of the BFlh at knee flexion angles >60° during leg curl [41].

For the MG and LG, the participants were instructed to stand on one leg and perform plantar flexion of the foot on each leg. For the GM, the participant was lying prone and lifting the entire leg against manual resistance. For the ES, the participants were asked to lift their trunks from a prone position against manual resistance. For the RA, the participants were asked to perform an abdominal curl against manual resistance. These MVIC positions were used in previous studies that investigated the EMG activity of the hamstring muscles [41,42]. Each MVIC protocol was performed for two bouts of 5 s. There was at least 1 minute of rest between each repetition. The EMG values were collected during each MVIC protocol. After the examiner recorded clean EMG readings of the target muscles during MVIC, the isometric NHex was used only once as a familiar attempt. After familiarization, isometric NHex was performed in bilateral and unilateral stances for 5 s under the following conditions as the randomized order: 1) 150° knee angle on a 50° inclined platform; and 2) 140° knee angle on a 40° inclined platform. For the isometric NHex protocol, participants were given instructions to begin sitting in a kneeling position at the designated degree with their elbows fully flexed and their hands positioned open in front of them. The examiner securely held the participant's ankle on the inclined platform while asking them to maintain a straight posture from the knees to the head. To monitor the knee angle, the examiner uses a manual goniometer during NHex. The external torque at the knee joint was equivalent for both tasks, as shown in Fig 1. A minimum of 1 minute of rest was given between each repetition and 2 minutes of rest between each NHex variation. The EMG data obtained under each condition were normalized to the values collected during the maximal voluntary isometric contraction of each muscle (normalized EMG [nEMG]). The experiment was conducted under the supervision of an examiner qualified as an Australian Strength and Conditioning Association Specialist.

## Data analysis

The EMG electrodes were pre-amplified (10×) and linked through the EMG mainframe, which further amplified it (100×) to a total gain of 1,000× and bandpass filter (20–450 Hz) signals. The sampling rate of the EMG was 2,000 Hz. The root mean square value during 2-s of 5-s MVICs was calculated as the mean. The root mean square value was calculated over a window of 100 ms. The percentage MVIC during the NHex variations was calculated by dividing the root mean square of each NHex variation by the mean value of the MVICs. The root mean squares of the EMG data obtained under each condition were normalized with the values collected during the maximal voluntary isometric contraction of each muscle (nEMG).

## Statistical analyses

The average value (+SD) of each exercise was calculated. Initially, the Shapiro–Wilk test was used to analyze normal distribution. Next, a two (between factor: non-HSI and previous history of HSI legs) by four (within factor: 40BL, 40UL, 50BL, 50UL) factor repeated-measure analysis of variance was used to confirm the main or interaction effects of NiEMG in each muscle and the ratio of BFlh/ST between groups. Greenhouse–Geisser corrections were applied when sphericity assumption was violated. When the main or interaction effects were significant, a Bonferroni post-hoc test was conducted to find the pairwise comparison in exercise variations. The partial $\eta^2$ was classified based on the following effect size criteria: trivial, <0.01; small, 0.01–0.06; medium, 0.06–0.14; and large, >0.14. The Cohen's d was classified based on the following effect size criteria: trivial, <0.2; small, 0.2–0.5; medium, 0.5–0.8; and large, >0.8. Significance level was set at $p < 0.05$. Statistical analyses were performed using JASP software (version 0.19.1).

## Results

### EMG activity of BFlh

For legs without a prior HSI injury, the main effects of the task (bilateral vs. unilateral) were significant (F = 23.47; partial $\eta^2$ = 0.72; $p < 0.01$). The graph on the left side of (Fig 2) displays the normalized integrated EMG activity (NiEMG) of the BFlh in the leg with no history of HSI. The EMG activity of the BFlh for UL40 and UL50 was higher than that for

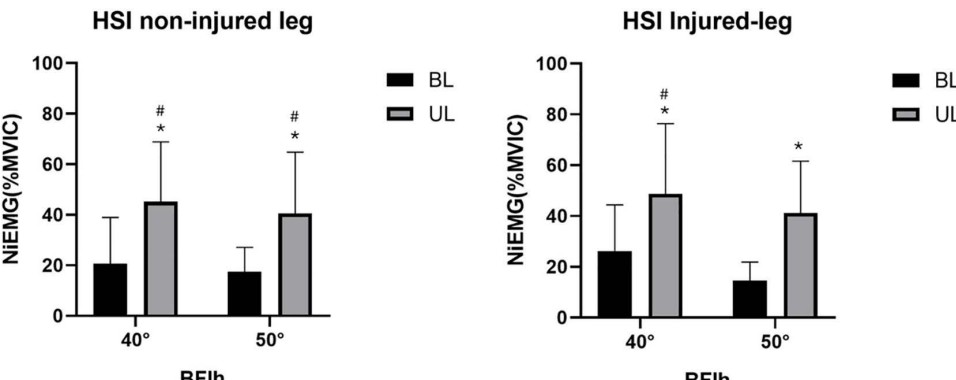

**Fig 2. Difference between the NiEMG (%MVIC) of the BFlh among 40BL, 40UL, 50BL, 50UL.** *Significant difference (p < 0.05) between BL and UL compared with the same slope degree. #Significant difference (p < 0.05) between BL and UL compared with the different slope degrees. 40BL: Bilateral Nordic hamstring exercise at 40° lower leg slope, 40UL: Unilateral Nordic hamstring exercise at 40° lower leg slope, 50BL: Bilateral Nordic hamstring exercise at 50° lower leg slope, 50UL: Unilateral Nordic hamstring exercise at 50° lower leg slope, MVIC maximum voluntary isometric contraction, BFlh: biceps femoris long head.

BL40 (d = 1.23; p = 0.009) and BL50 (d = 1.16; p = 0.027). Furthermore, UL40 was significantly greater than BL50 (d = 1.40; p = 0.025), and UL50 showed a greater value than BL40 (d = 1.00; p = 0.025). For legs with a previous history of HSI (the right side of Fig 2), the main effect of task (bilateral vs. unilateral task) factors was significant (F = 19.795; partial η2 = 0.460; p = 0.002). The EMG activity of the BFlh for UL40 and UL50 was higher than that for BL40 (d = 1.13; p = 0.036) and BL50 (d = 1.34; p = 0.006). Furthermore, UL40 expression was significantly higher than BL50 expression (d = 1.72; p = 0.012).

## EMG activity of BFlh/ST ratio

For legs without a prior HSI injury, the main effects of the task (bilateral vs. unilateral) were significant (F = 11.86; partial η2 = 0.59; p = 0.009). The graph on the left side of Fig 3 displays the EMG activity of the BFlh/ST ratio in the leg with no history of HSI. The EMG activity of the BFlh/ST ratio for UL50 was higher than that for BL50 (d = −0.75; p = 0.020). Additionally, UL50 was greater than BL40 (d = 0.98; p = 0.037). The graph on the right side of Fig 3 shows the EMG activity of the BFlh/ST ratio in a leg with a history of HSI. The EMG activity of the BFlh/ST ratio at UL50 was higher than that at UL40 (d = 0.75; p = 0.020). Additionally, UL50 was greater than BL40 (d = 0.98; p = 0.037). For legs with a previous history of HSI, the main effect of task (bilateral vs. unilateral task) factors was significant (F = 29.614; partial η2 = 0.809; p < 0.001). The EMG activity of the BFlh/ST ratio for UL40 and UL50 was higher than that for BL40 (d = 1.71; p = 0.015) and BL50 (d = 2.32; p = 0.016). Furthermore, UL50 was significantly greater than BL40 (d = 1.62; p = 0.016), and UL40 showed a greater value than BL50 (d = 2.41; p = 0.013).

## EMG activity related muscles

Table 1 shows the EMG activity of related muscles (ES, GM, RA, LA, and MG) in the legs with and without previous HSI. There was a significant difference only between the GM and MG groups. For the GM, the main effect of task factors (BL and UL) was significant (F = 11.20; partial η2 = 0.61; p = 0.012) and slope factors (40° and 50°) was significant (F = 8.37; partial η2 = 0.54; p = 0.023) in non-previous HSI side. A slope of 40° was larger than that at 50° (d = −0.918; p = 0.035) and UL was higher than BL (d = 0.50; p = 0.026). For the previous HSI injured leg, the main effect of task factors (BL and UL) was significant (F = 17.54; partial η2 = 0.71; p = 0.004), and slope factors (40° and 50°) was significant (F = 9.20; partial η2 = 0.56; p = 0.019). Additionally, 40UL was higher than 50 BL (d = 2.22; p = 0.019) and 40UL was higher than 40 BL (d = 2.16; p = 0.017).

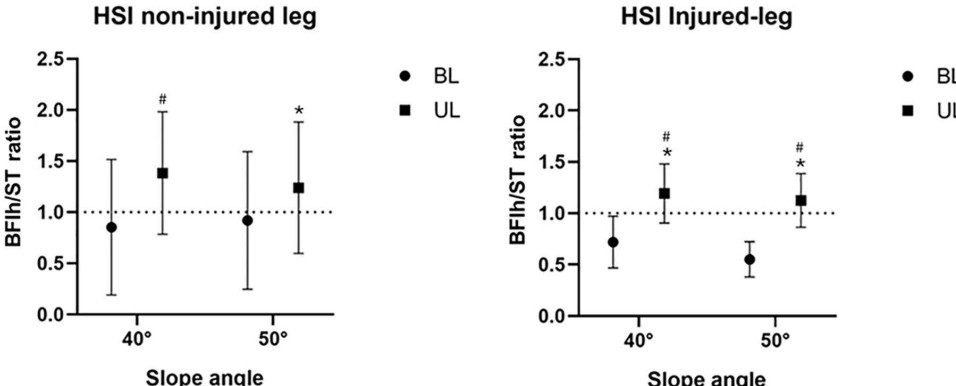

**Fig 3. Difference of the NiEMG (%MVIC) of BFlh/ST ratio among 40BL, 40UL, 50BL, 50UL.** *Significant difference (p<0.05) between BL and UL compared with the same degree. # Significant difference (p<0.05) between BL and UL compared with different degrees. 40BL: Bilateral Nordic hamstring exercise at 40° lower leg slope, 40UL: Unilateral Nordic hamstring exercise at 40° lower leg slope, 50BL: Bilateral Nordic hamstring exercise at 50° lower leg slope, 50UL: Unilateral Nordic hamstring exercise at 50° lower leg slope, MVIC maximum voluntary isometric contraction, BFlh: biceps femoris long head, ST: semitendinosus. The ratio over 1.0 means that the NiEMG of BF is higher than that of ST (solid horizontal line).

**Table 1. EMG activity of the ES, GM, RA, LG, and MG muscle during NHex variations.**

| Related muscles | NHex50 BL | NHex50 UL (NI) | NHex50UL (I) | NHex40 BL | NHex40 UL (NI) | NHex40UL (I) |
|---|---|---|---|---|---|---|
| ES (NI) | 12.43±5.02 | 15.73±8.47 | 20.42±10.09 | 18.66±9.00 | 18.33±10.77 | 27.06±13.59 |
| ES (I) | 16.65±3.27 | 23.06±10.00 | 12.66±8.44 | 21.59±11.20 | 26.24±10.59 | 20.04±11.61 |
| GM (NI) | 1.51±0.96 | 4.92±4.49 | 1.10±0.50 | 2.05±1.13 | 4.55±3.01 | 2.21±1.23 |
| GM (I) | 2.01±1.10 | 2.63±1.70 | 4.51±2.74 | 2.32±1.41 | 2.91±1.98 | 8.40±4.52*# |
| RA (NI) | 1.78±1.04 | 2.93±2.13 | 2.44±1.48 | 1.30±0.58 | 3.31±2.07 | 5.40±4.89 |
| RA (I) | 0.92±0.30 | 1.56±1.09 | 1.33±0.52 | 0.91±0.36 | 1.89±0.87 | 2.29±1.19 |
| LG (NI) | 11.57±8.17 | 12.34±5.04 | 6.89±7.19 | 20.56±19.48 | 43.29±30.45 | 3.33±3.00 |
| LG (I) | 16.54±8.63 | 7.11±8.49 | 46.53±57.49 | 18.39±12.43 | 4.46±3.63 | 43.04±25.83 |
| MG (NI) | 28.18±10.02 | 57.70±25.69 | 8.12±9.56 | 34.44±18.63 | 59.57±33.78 | 4.78±5.12 |
| MG (I) | 25.45±6.01 | 10.36±10.87 | 48.30±25.91 | 26.78±23.65 | 5.09±3.84 | 46.35±24.12 |

The NiEMG (%MVIC) of the ES, GM, RA, LG, and MG among 40BL, 40UL, 50BL, and 50UL. 40BL: Bilateral Nordic hamstring exercise at 40° lower leg slope, 40UL: Unilateral Nordic hamstring exercise at 40° lower leg slope, 50BL: Bilateral Nordic hamstring exercise at 50° lower leg slope, 50UL: Unilateral Nordic hamstring exercise at 50° lower leg slope, MVIC maximum voluntary isometric contraction, ES: erector spinae, GM:gluteus maximus, RA: rectus abdominis, LA:lateral gastrocnemius, MG: medial gastrocnemius, NI: Non-injured leg, I: injured leg.

*Significant difference (p < 0.05) compared to slope angles.

#Significant difference (p < 0.05) compared with the task

For MG, the main effect of slope factors (40° and 50°) was significant (F = 15.26; partial η2 = 0.68; p = 0.006) only in the non-previous HSI leg. The UL was higher than the BL (d = 1.03; p = 0.06).

## Discussion

This study examined the EMG activity of the hamstrings and related muscles during variations in isometric NHex performed on an inclined platform. The primary finding of this study was that the EMG activity of the BFlh was predominantly recruited during the unilateral NHex tasks, regardless of whether the leg had HSI or not. In contrast, the EMG activities of the BFlh and ST muscles were nearly equal during the bilateral NHex at shallow knee flexion angles on the inclined platform, particularly in the non-previous HSI-injured leg. Additionally, the study found no significant differences

in EMG activity across various slope angles, even when the external torques at the knee joint were estimated to be equivalent.

This study demonstrated that the isometric unilateral NHex with sloped platforms showed significantly higher BFlh EMG activity than the isometric bilateral NHex alone. Additionally, this study also found that BFlh EMG activity was higher than ST activity only in unilateral NHex conditions. To the best of our knowledge, no previous study has demonstrated that the NiEMG of the BFlh can be recruited during isometric NHex. However, Soga et al. examined NHex during an eccentric contraction and reported that BFlh EMG activity was significantly higher than that of the ST in unilateral NHex with a sloped platform (Soga et al., 2022). As speculated in previous studies, one possible mechanism is that unilateral exercises yield greater force per limb than bilateral exercises [43,44]. Additionally, the amount of hamstring force produced has been proposed to be related to hamstring activity patterns [43,45]. Regarding a previous study, the larger physiological cross-sectional area of the BFlh compared with the ST was beneficial for force production [46], leading to an assumption that the BFlh may preferentially engage during the unilateral NHex, even during isometric contraction, which necessitates significant force generation. Additionally, our findings were in line with those of Hirose et al. (Hirose et al., 2021); they reported that the BFlh and ST muscles were equally activated during bilateral NHex at shallow knee flexion angles with an inclination platform, especially in the non-previous HSI leg. Numerous studies have highlighted the limitations of the NHex due to its failure to activate the BFlh muscle, which is particularly vulnerable to HSI [36,47]. The current study indicates that the NHex may be effective for HSI prevention [3,8,48–50] when performed with the knee positioned at a shallow flexion angle [35] and unilateral stance. Although statistical differences were not examined, it was observed that during the bilateral NHex, the EMG activities of the BFlh and ST were nearly equal in the HSI-non-injured leg. In contrast, in the HSI-injured leg, the BFlh/ST ratio was significantly lower than 1.0, indicating greater ST activity than BFlh activity. This finding suggests that even when performing the NHex with a shallow knee flexion angle using an inclined platform, individuals with a history of HSI may have difficulty activating the BFlh muscle efficiently. This underscores the clinical significance of our findings, highlighting the need to adjust the platform angle according to the recovery status of the injured leg and implement unilateral NHex accordingly.

This study found no differences in the BFlh and BFlh/ST ratio EMG activity when the slope and knee flexion angles were set differently. However, the external torque at the knee joint was calculated to be equivalent between tasks by modulating the knee flexion angle. Hirose et al. [35] reported that performing the NHex at a shallow knee flexion angle may assist athletes in performing eccentric exercises in lengthened positions, activating the BFlh muscle to the same extent as the ST muscles compared with conventional NHex, which is in line with our findings. Moreover, our findings showed that alterations in the length of hamstring muscles between 40–50° slope tasks may not be sufficient to modulate hamstring activity, which is in line with the study by Hirose et al. [35]. This previous study reported that there was no significant difference in the relative EMG activity of the BFlh and ST muscles by increasing a slope angle from 50° to 40° (Hirose et al., 2021). This study does not provide a clear underlying mechanism for this result. As speculated in a previous study [41], increasing external loads (manual resistance from 20% to 40% MVIC) enhanced the EMG activity in the BFlh and ST muscles during the prone leg curl. From our findings, performing NHex at a shallower knee flexion angle of 40–50° by setting equivalent external torque may not affect hamstring activity.

For the related muscle, this study found that GM was more recruited during unilateral stance and at a 40° slope angle only in previous HSI injured leg. Our findings align with those of previous studies, indicating that the Gmax activity diminishes when the knee extension angle increases. In other words, when the knee extension angle increases, the demand for ES activity exceeds that of the GM [51]. The GM and hamstring muscles may produce internal hip extension torque to maintain the alignment of the pelvis and upper trunk during the NHex, while the ES also stabilizes the upper body during the NHex (Narouei et al., 2018). If each Gmax or ES muscle engages inefficiently, hamstring activity may increase as a compensatory mechanism (Narouei et al., 2018). It may be that Gmax may compensate for hamstring activity to keep the pelvis and upper trunk aligned in the previous HSI leg.

This study has several limitations. This study did not measure force; instead, it estimated that the external torque at the knee joint could be determined by calculations based on previous studies [52].This study did not measure the knee flexion angle using electrogoniometer. Although manual goniometers produce highly reliable results [24,53], additional studies are required on digital measurements. Moreover, this study specifically recruited male volunteers with a history of HSI within the past two years. The results may vary based on other participant factors (e.g., gender or athletic status). Despite these limitations, the findings of this study have practical applications. For example, trained athletes are unable to perform the unilateral NHex at shallow knee flexion angles [35,39,54]. In this regard, the use of an inclined platform can help athletes perform the unilateral NHex at shallow knee flexion angles. However, to implement this exercise in a real-world training setting, the incline platform is necessarily required. To overcome this challenge, this study recommends an assisted exercise band that may help to keep the trunk at the shallow knee flexion angle. In our study, the unilateral NHex preferentially recruited BFlh in comparison with the bilateral NHex on the inclined platform, which has positive effects in preventing HSI.

## Conclusions

This study showed that the BFlh was primarily engaged during isometric unilateral NHex tasks, irrespective of the leg's history of HSI, while both the BFlh and ST muscles exhibited nearly comparable activation during the NHex at shallow knee flexion angles on the inclined platform, particularly in the non-injured leg. Furthermore, variations in the length of hamstring muscles during tasks on a 40° to 50° slope may be inadequate to influence hamstring activity despite the same external torque at the knee joint. A recent study emphasized the importance of recruiting BFlh to assist healthcare professionals in developing a progressive program aimed at activating the BFlh muscle for injury prevention and rehabilitation strategies [33]. Based on these data, we deduce that executing the isometric unilateral NHex at a shallow knee flexion angle on an inclined platform may be the preferable hamstring exercise by engaging the BFlh muscle which is susceptible to HSI.

## Supporting information

**S1 Dataset. Comparing the hamstring muscle activity between injured and non-injured sides during a variety of Nordic hamstring exercises.**
(XLSX)

**S1 File. Inclusivity-in-global-research-questionnaire.**
(DOCX)

## Acknowledgments

The authors would like to thank Mr. James and the students for assisting with data collection and to acknowledge the facilities and the assistance of College of Sports Science and Technology, Mahidol University.

## Author contributions

**Conceptualization:** Weerawat Limroongreungrat, Norikazu Hirose.

**Data curation:** Taspol Keerasomboon.

**Formal analysis:** Taspol Keerasomboon, Toshiaki Soga.

**Investigation:** Taspol Keerasomboon.

**Methodology:** Taspol Keerasomboon, Parunchaya Jamkrajang, Thammanunt Chrunarm, Norikazu Hirose.

**Resources:** Taspol Keerasomboon.

**Software:** Taspol Keerasomboon.

**Supervision:** Norikazu Hirose.

**Writing – original draft:** Taspol Keerasomboon.

**Writing – review & editing:** Taspol Keerasomboon, Toshiaki Soga.

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
