## [Decision Letter · Decision Letter 0]

8 Apr 2025

PONE-D-25-13241Comparing the hamstring muscle activity between injured and non-injured sides during a variety of Nordic hamstring exercisesPLOS ONE

Dear Dr. Hirose,

Thank you for submitting your manuscript to PLOS ONE. After careful consideration, we feel that it has merit but does not fully meet PLOS ONE’s publication criteria as it currently stands. Therefore, we invite you to submit a revised version of the manuscript that addresses the points raised during the review process.

We look forward to receiving your revised manuscript.

Kind regards,

Prateek Srivastav

Academic Editor

PLOS ONE

Journal Requirements:

4. We notice that your supplementary figures are uploaded with the file type 'Figure'. Please amend the file type to 'Supporting Information'. Please ensure that each Supporting Information file has a legend listed in the manuscript after the references list.

5. We notice that your supplementary table is included in the manuscript file. Please remove them and upload them with the file type 'Supporting Information'. Please ensure that each Supporting Information file has a legend listed in the manuscript after the references list.

Reviewers' comments:

Reviewer's Responses to Questions

**Comments to the Author**

1. Is the manuscript technically sound, and do the data support the conclusions?

Reviewer #1: Partly

Reviewer #2: Yes

2. Has the statistical analysis been performed appropriately and rigorously?

Reviewer #1: Yes

Reviewer #2: Yes

3. Have the authors made all data underlying the findings in their manuscript fully available?

Reviewer #1: Yes

Reviewer #2: Yes

4. Is the manuscript presented in an intelligible fashion and written in standard English?

Reviewer #1: Yes

Reviewer #2: Yes

5. Review Comments to the Author

Reviewer #1: In this manuscript, the authors present preliminary results that could serve as a basis for studying new treatment/prevention strategies for hamstring injuries. Despite the interest and possible impact of the study, some aspects of the manuscript need to be revised in order to improve its quality:

ABSTRACT

Line 51-53- ‘In contrast, electromyographic activity of the Biceps femoris long head and Semitendinosus muscles was approximately equivalent in non injured legs without a history of hamstring strain injury.’ Wouldn't it be more relevant to replace this information with ‘BFlh EMG activity was higher than ST activity only in unilateral NHex conditions’?

INTRODUCTION

This section needs to be revised in order to summarise the information and improve the flow of ideas.

METHODS

Line 153/154- The inclusion criteria state that ‘NHex was not included in the regular resistance training programme. ‘ However, the inclusion criteria do not include resistance training.

Discussion

The impact of the manuscript and the applicability of its results would benefit from exploring the implications for rehabilitation/injury prevention in greater detail.

Limitations

A compromise in external validity needs to be recognised due to the particular characteristics of the participants (volunteers with a history of HSI within two years). Results may be different in athletes ???

Translated with DeepL.com (free version)

Reviewer #2: Title and Abstract

• Clarity and Relevance: The title is clear and directly relevant to the content of the paper. The abstract effectively summarizes the study's objectives, methodology, and key findings, highlighting the comparison of electromyographic activity in hamstring muscles between injured and non-injured sides during different Nordic hamstring exercises.

• Improvement Suggestion: Consider adding more specific details about the implications of these findings for injury prevention and rehabilitation strategies.

Introduction

• Depth and Context: The introduction provides a comprehensive background on hamstring strain injuries (HSI), their prevalence, recurrence rates, and the importance of preventing recurrence. It sets a clear context for the study's focus on comparing muscle activity during different exercises.

• Improvement Suggestion: Include more specific details about the current state of research on muscle activity in injured vs. non-injured hamstrings to further contextualize the study's objectives.

Literature Review

• Comprehensiveness: The literature review is thorough, covering various aspects of HSI, including risk factors, recurrence rates, and the role of muscle structure and function. However, it could benefit from a more detailed discussion on the specific benefits of Nordic hamstring exercises for injury prevention.

• Improvement Suggestion: Expand on the literature regarding the effectiveness of different exercise modalities in reducing HSI recurrence.

Methodology

• Appropriateness and Clarity: The methodology is well-described, using a robust approach to compare electromyographic activity during bilateral and unilateral isometric Nordic hamstring exercises. The control of external torque at the knee joint enhances the study's rigor.

• Rigor: The inclusion of a small but focused sample of physically active males with a history of HSI and the use of electromyography to measure muscle activity add to the study's reliability.

• Improvement Suggestion: Consider providing more details on the criteria used for selecting participants with a history of HSI.

Results Presentation, Analysis, and Interpretation

• Presentation: The results are clearly presented, with tables and figures supporting the text. The analysis is robust, using appropriate statistical methods to evaluate muscle activity differences.

• Interpretation: The findings are well-interpreted, highlighting significantly higher electromyographic activity of the Biceps femoris long head during unilateral exercises. However, more emphasis could be placed on the practical implications for training programs.

• Improvement Suggestion: Include more detailed explanations for the observed differences in muscle activity between bilateral and unilateral exercises.

Discussion

• Relation to Research Questions and Prior Literature: The discussion effectively links the findings to the research questions and prior literature, emphasizing the potential benefits of unilateral exercises for enhancing muscle activation.

• Limitations and Implications: The study acknowledges limitations and highlights implications for training strategies, which is commendable. However, more emphasis could be placed on potential future studies to validate these findings in larger populations.

• Improvement Suggestion: Expand on potential barriers to implementing unilateral exercises in real-world training settings and suggest strategies to overcome these challenges.

Conclusion

• Summary of Key Findings: The conclusion succinctly summarizes the key findings, emphasizing the benefits of unilateral isometric Nordic hamstring exercises for enhancing Biceps femoris long head muscle activity.

• Contributions and Recommendations: It highlights the contributions of the study and provides recommendations for future research and practice.

• Improvement Suggestion: Consider adding a statement on the potential for this research to inform coaching practices or rehabilitation protocols.

References

• Relevance, Accuracy, and Formatting: The references appear relevant and accurately formatted according to the chosen citation style.

• Improvement Suggestion: Ensure that all references are up-to-date and include a mix of recent and foundational studies.

Writing Quality, Organization, and Readability

• The paper is well-organized and readable, with clear headings and concise paragraphs. However, some sections could benefit from more transitional phrases to enhance flow.

• Improvement Suggestion: Use more active voice in certain sections to improve clarity and engagement.

Originality, Contribution, and Ethical Considerations

• Originality and Contribution: The study contributes significantly by highlighting the benefits of unilateral Nordic hamstring exercises for muscle activation, providing valuable insights for injury prevention and rehabilitation.

• Ethical Considerations: The study appears to adhere to ethical standards, as it involves human subjects with appropriate institutional review board approval.

• Improvement Suggestion: Consider discussing broader ethical implications of promoting specific exercise modalities, such as ensuring accessibility for all athletes.

6. PLOS authors have the option to publish the peer review history of their article (what does this mean? ). If published, this will include your full peer review and any attached files.

**Do you want your identity to be public for this peer review?** For information about this choice, including consent withdrawal, please see our Privacy Policy .

Reviewer #1: No

Reviewer #2: **Yes: ** Yaser Alikhajeh

---

## [Author Response · Author response to Decision Letter 1]

22 Apr 2025

Revisions and answers made in response to the reviewer's comments

Review Comments to the Author

Reviewer #1: In this manuscript, the authors present preliminary results that could serve as a basis for studying new treatment/prevention strategies for hamstring injuries. Despite the interest and possible impact of the study, some aspects of the manuscript need to be revised in order to improve its quality:

Response to the reviewer's comments: Thank you very much for your suggestion.

ABSTRACT

Line 51-53- ‘In contrast, electromyographic activity of the Biceps femoris long head and Semitendinosus muscles was approximately equivalent in non injured legs without a history of hamstring strain injury.’ Wouldn't it be more relevant to replace this information with ‘BFlh EMG activity was higher than ST activity only in unilateral NHex conditions’?

Response to the reviewer's comments: Thank you very much for your suggestion. We agree with your suggestion. We have already revised the sentence following your suggestion. (line 51-53)

INTRODUCTION

This section needs to be revised in order to summarise the information and improve the flow of ideas.

Response to the reviewer's comments: Thank you very much for your suggestion. We have already revised the sentence to improve the flow of ideas. (98-102, 112-114)

METHODS

Line 153/154- The inclusion criteria state that ‘NHex was not included in the regular resistance training programme. ‘ However, the inclusion criteria do not include resistance training.

Response to the reviewer's comments: Thank you very much for your question and suggestion. We want to point out that the participants have not before performed the Nordic hamstring exercise. We have revised the sentence to “the NHex was not previously performed.” (line 164-165)

Discussion

The impact of the manuscript and the applicability of its results would benefit from exploring the implications for rehabilitation/injury prevention in greater detail.

Response to the reviewer's comments: Thank you very much for your suggestion.

Limitations

A compromise in external validity needs to be recognised due to the particular characteristics of the participants (volunteers with a history of HSI within two years). Results may be different in athletes ???

Response to the reviewer's comments: Thank you very much for your suggestion. We have already revised following your suggestion by adding the limitation about characteristics of the participants (line 365-367)

Translated with DeepL.com (free version)

Reviewer #2: Title and Abstract

• Clarity and Relevance: The title is clear and directly relevant to the content of the paper. The abstract effectively summarizes the study's objectives, methodology, and key findings, highlighting the comparison of electromyographic activity in hamstring muscles between injured and non-injured sides during different Nordic hamstring exercises.

Response to the reviewer's comments: Thank you very much for your comment.

• Improvement Suggestion: Consider adding more specific details about the implications of these findings for injury prevention and rehabilitation strategies. (line 383-386)

Response to the reviewer's comments: Thank you very much for your suggestions. I have already added more specific details about the implications of these findings for injury prevention and rehabilitation strategies. (line 383-386)

Introduction

• Depth and Context: The introduction provides a comprehensive background on hamstring strain injuries (HSI), their prevalence, recurrence rates, and the importance of preventing recurrence. It sets a clear context for the study's focus on comparing muscle activity during different exercises.

Response to the reviewer's comments: Thank you very much for your comment.

• Improvement Suggestion: Include more specific details about the current state of research on muscle activity in injured vs. non-injured hamstrings to further contextualize the study's objectives.

Response to the reviewer's comments: Thank you very much for your suggestions. We have already added the details following your suggestion by providing more specific details regarding the current research on muscle activity in injured vs. non-injured hamstrings. (line 98-102)

Literature Review

• Comprehensiveness: The literature review is thorough, covering various aspects of HSI, including risk factors, recurrence rates, and the role of muscle structure and function. However, it could benefit from a more detailed discussion on the specific benefits of Nordic hamstring exercises for injury prevention.

Response to the reviewer's comments: Thank you very much for your comment.

• Improvement Suggestion: Expand on the literature regarding the effectiveness of different exercise modalities in reducing HSI recurrence.

Response to the reviewer's comments: Thank you very much for your suggestions. We have already added the details following your suggestion by providing more specific details regarding the effectiveness of different exercise modalities in reducing HSI recurrence. (line 112-114)

Methodology

• Appropriateness and Clarity: The methodology is well-described, using a robust approach to compare electromyographic activity during bilateral and unilateral isometric Nordic hamstring exercises. The control of external torque at the knee joint enhances the study's rigor.

Response to the reviewer's comments: Thank you very much for your comment.

• Rigor: The inclusion of a small but focused sample of physically active males with a history of HSI and the use of electromyography to measure muscle activity add to the study's reliability.

Response to the reviewer's comments: Thank you very much for your comment.

• Improvement Suggestion: Consider providing more details on the criteria used for selecting participants with a history of HSI.

Response to the reviewer's comments: Thank you very much for your suggestion. We have already added the details following your suggestion regarding the criteria used for selecting participants with a history of HSI. (line 159-161)

Results Presentation, Analysis, and Interpretation

• Presentation: The results are clearly presented, with tables and figures supporting the text. The analysis is robust, using appropriate statistical methods to evaluate muscle activity differences.

Response to the reviewer's comments: Thank you very much for your comment.

• Interpretation: The findings are well-interpreted, highlighting significantly higher electromyographic activity of the Biceps femoris long head during unilateral exercises. However, more emphasis could be placed on the practical implications for training programs.

Response to the reviewer's comments: Thank you very much for your comment.

• Improvement Suggestion: Include more detailed explanations for the observed differences in muscle activity between bilateral and unilateral exercises.

Response to the reviewer's comments: Thank you very much for your suggestion.

Discussion

• Relation to Research Questions and Prior Literature: The discussion effectively links the findings to the research questions and prior literature, emphasizing the potential benefits of unilateral exercises for enhancing muscle activation.

Response to the reviewer's comments: Thank you very much for your comment.

• Limitations and Implications: The study acknowledges limitations and highlights implications for training strategies, which is commendable. However, more emphasis could be placed on potential future studies to validate these findings in larger populations.

Response to the reviewer's comments: Thank you very much for your suggestion.

• Improvement Suggestion: Expand on potential barriers to implementing unilateral exercises in real-world training settings and suggest strategies to overcome these challenges.

Response to the reviewer's comments: Thank you very much for your suggestion. We have already added the details following your suggestion about potential barriers to implementing unilateral exercises in real-world training settings and suggest strategies to overcome these challenges. (line 371-374)

Conclusion

• Summary of Key Findings: The conclusion succinctly summarizes the key findings, emphasizing the benefits of unilateral isometric Nordic hamstring exercises for enhancing Biceps femoris long head muscle activity.

Response to the reviewer's comments: Thank you very much for your comment.

• Contributions and Recommendations: It highlights the contributions of the study and provides recommendations for future research and practice.

Response to the reviewer's comments: Thank you very much for your comment.

• Improvement Suggestion: Consider adding a statement on the potential for this research to inform coaching practices or rehabilitation protocols.

Response to the reviewer's comments: Thank you very much for your suggestion. We have already added the details following your suggestion about the potential for this research to inform coaching practices or rehabilitation protocols (line 383-386)

References

• Relevance, Accuracy, and Formatting: The references appear relevant and accurately formatted according to the chosen citation style.

Response to the reviewer's comments: Thank you very much for your comment.

• Improvement Suggestion: Ensure that all references are up-to-date and include a mix of recent and foundational studies.

Response to the reviewer's comments: Thank you very much for your suggestion.

Writing Quality, Organization, and Readability

• The paper is well-organized and readable, with clear headings and concise paragraphs. However, some sections could benefit from more transitional phrases to enhance flow.

Response to the reviewer's comments: Thank you very much for your comment.

• Improvement Suggestion: Use more active voice in certain sections to improve clarity and engagement.

Response to the reviewer's comments: Thank you very much for your suggestion.

Originality, Contribution, and Ethical Considerations

• Originality and Contribution: The study contributes significantly by highlighting the benefits of unilateral Nordic hamstring exercises for muscle activation, providing valuable insights for injury prevention and rehabilitation.

• Ethical Considerations: The study appears to adhere to ethical standards, as it involves human subjects with appropriate institutional review board approval.

Response to the reviewer's comments: Thank you very much for your comment.

• Improvement Suggestion: Consider discussing broader ethical implications of promoting specific exercise modalities, such as ensuring accessibility for all athletes.

Response to the reviewer's comments: Thank you very much for your suggestion.

---

## [Decision Letter · Decision Letter 1]

13 May 2025

Comparing the hamstring muscle activity between injured and non-injured sides during a variety of Nordic hamstring exercises

PONE-D-25-13241R1

Dear Dr. Hirose,

We’re pleased to inform you that your manuscript has been judged scientifically suitable for publication and will be formally accepted for publication once it meets all outstanding technical requirements.

Kind regards,

Prateek Srivastav

Academic Editor

PLOS ONE

Reviewers' comments:

Reviewer's Responses to Questions

**Comments to the Author**

1. If the authors have adequately addressed your comments raised in a previous round of review and you feel that this manuscript is now acceptable for publication, you may indicate that here to bypass the “Comments to the Author” section, enter your conflict of interest statement in the “Confidential to Editor” section, and submit your "Accept" recommendation.

Reviewer #1: All comments have been addressed

Reviewer #2: (No Response)

2. Is the manuscript technically sound, and do the data support the conclusions?

Reviewer #1: Yes

Reviewer #2: Yes

3. Has the statistical analysis been performed appropriately and rigorously?

Reviewer #1: Yes

Reviewer #2: Yes

4. Have the authors made all data underlying the findings in their manuscript fully available?

Reviewer #1: Yes

Reviewer #2: Yes

5. Is the manuscript presented in an intelligible fashion and written in standard English?

Reviewer #1: Yes

Reviewer #2: Yes

6. Review Comments to the Author

Reviewer #1: (No Response)

Reviewer #2: Thanks for responding to all the comments. As a final suggestion, please review the article for spelling, for example, on line 53.

7. PLOS authors have the option to publish the peer review history of their article (what does this mean? ). If published, this will include your full peer review and any attached files.

**Do you want your identity to be public for this peer review?** For information about this choice, including consent withdrawal, please see our Privacy Policy .

Reviewer #1: No

Reviewer #2: **Yes: ** Yaser Alikhajeh

---

## [Editor Report · Acceptance letter]

PONE-D-25-13241R1

PLOS ONE

Dear Dr. Hirose,

I'm pleased to inform you that your manuscript has been deemed suitable for publication in PLOS ONE. Congratulations! Your manuscript is now being handed over to our production team.

Kind regards,

on behalf of

Dr. Prateek Srivastav

Academic Editor

PLOS ONE